# Outcomes of Cancer Patients with COVID-19 in a Hospital System in the Chicago Metropolitan Area

**DOI:** 10.3390/cancers14092209

**Published:** 2022-04-28

**Authors:** Alain Mina, Carlos Galvez, Reem Karmali, Mary Mulcahy, Xinlei Mi, Masha Kocherginsky, Michael J Gurley, Neelima Katam, William Gradishar, Jessica K Altman, Michael G Ison, Dean Tsarwhas, Christopher George, Jane N Winter, Leo I. Gordon, Firas H Wehbe, Leonidas C Platanias

**Affiliations:** 1Division of Blood and Marrow Transplantation and Cellular Therapies, Department of Medicine, Stanford University School of Medicine, Stanford, CA 94305, USA; 2Robert H. Lurie Comprehensive Cancer Center, Northwestern University, Chicago, IL 60611, USA; carlos.galvez@northwestern.edu (C.G.); reem.karmali@northwestern.edu (R.K.); m-mulcahy@northwestern.edu (M.M.); xinlei.mi@northwestern.edu (X.M.); mkocherg@northwestern.edu (M.K.); m-gurley@northwestern.edu (M.J.G.); neelima.katam@northwestern.edu (N.K.); w-gradishar@northwestern.edu (W.G.); j-altman@northwestern.edu (J.K.A.); dean.tsarwhas@nm.org (D.T.); christopher.george@northwestern.edu (C.G.); j-winter@northwestern.edu (J.N.W.); l-gordon@northwestern.edu (L.I.G.); firas.wehbe@northwestern.edu (F.H.W.); l-platanias@northwestern.edu (L.C.P.); 3Division of Hematology-Oncology, Department of Medicine, Feinberg School of Medicine, Northwestern University, Chicago, IL 60611, USA; 4Divisions of Infectious Diseases and Organ Transplantation, Feinberg School of Medicine, Northwestern University, Chicago, IL 60611, USA; mgison@northwestern.edu; 5Northwestern Medicine Lake Forest Hospital, Lake Forest, IL 60045, USA; 6Northwestern Medicine Central DuPage Hospital, Winfield, IL 60190, USA; 7Northwestern Medicine Delnor Hospital, Geneva, IL 60134, USA

**Keywords:** cancer, SARS-CoV-2, COVID-19

## Abstract

**Simple Summary:**

The spectrum of COVID-19 clinical presentation is wide and ranges from a mild flu-like illness to severe life-threatening respiratory illness with multiorgan involvement. The effects of the pandemic were particularly serious in patients with a history of malignancy, particularly those who are undergoing certain anti-cancer therapies. Our manuscript summarizes our experience with COVID-19 in our cancer program in Chicago, IL and investigates the different clinical and laboratory parameters that impact severity of outcomes and 30-day mortality in our cancer patients.

**Abstract:**

Patients with a history of malignancy have been shown to be at an increased risk of COVID-19-related morbidity and mortality. Poorer clinical outcomes in that patient population are likely due to the underlying systemic illness, comorbidities, and the cytotoxic and immunosuppressive anti-tumor treatments they are subjected to. We identified 416 cancer patients with SARS-CoV-2 infection being managed for their malignancy at Northwestern Medicine in Chicago, Illinois, between March and July of 2020. Seventy-five (18.0%) patients died due to COVID-related complications. Older age (>60), male gender, and current treatment with immunotherapy were associated with shorter overall survival. Laboratory findings showed that higher platelet counts, ALC, and hemoglobin were protective against critical illness and death from COVID-19. Conversely, elevated inflammatory markers such as ferritin, d-dimer, procalcitonin, CRP, and LDH led to worse clinical outcomes. Our findings suggest that a thorough clinical and laboratory assessment of infected patients with cancer might help identify a more vulnerable population and implement more aggressive proactive strategies.

## 1. Introduction

Since January 2020, the global COVID-19 pandemic has resulted in more than 258 million infections and 5 million deaths worldwide [1]. The clinical spectrum of COVID-19 ranges from a mild flu-like illness to severe life threatening multiorgan involvement [2]. The leading cause of death appears to be viral pneumonia causing acute respiratory distress syndrome (ARDS) [3]. Most studies have linked male gender, advanced age, and the presence of comorbidities, including obesity, lung disease, cardiovascular disease, and hypertension, to more severe infections and poorer clinical outcomes [4,5,6,7,8,9,10,11]. Data are less robust when it comes to infection rates and clinical outcomes in patients with a history of malignancy. Cancer patients are a widely heterogenous population with regards to the type of malignancy, extent of disease, treatment history, and baseline performance status. They are considered a vulnerable population in the context of the COVID-19 pandemic in large part because of the increased metabolic and inflammatory stress of a systemic malignancy, and the cytotoxic and immunosuppressive anti-tumor treatments needed [12,13]. They often also have other comorbid conditions that contribute to risk of severe disease. In a retrospective analysis of 28 cancer patients with COVID-19, Zhang et al. demonstrated that patients who had undergone anti-cancer therapy within two weeks of SARS-CoV-2 infection were more likely to develop severe COVID-19 illnesses [14]. Similarly, Mehta et al. reported the experience of a single medical center in New York City, which found that the COVID-19 related mortality of 218 cancer patients was 28%, almost three-fold greater than their non-cancer counterparts [15]. The COVID-19 and Cancer Consortium (CCC19) database retrospectively looked at the baseline clinical data of a cohort of 928 patients with active or previous malignancy. Data generated established that patients with a history of cancer had a more severe COVID-19 illness and a higher fatality rate compared to that of the general population [16]. Patients with malignancy are also more likely to get infected with SARS-CoV-2 owing to the nosocomial transmission of COVID-19 and the need of cancer patients for frequent hospital visits and regular contact with virus-contaminated areas [17,18].

Given the global impact of the COVID-19 pandemic and the increased prevalence of cancer in the general population, particularly in patients with advanced age and multiple comorbidities, better understanding of the clinical course of COVID-19 in the cancer population has become an urgent necessity. We therefore sought to summarize our experience with COVID-19 in our cancer program and investigate the different clinical parameters that impact severity of outcomes and 30-day mortality. The goal was to predict clinical course and tailor a plan of action based on clinical and laboratory parameters.

## 2. Materials and Methods

### 2.1. Study Design

This is a retrospective cohort analysis of patients who had tested positive for the COVID-19 infection from March 2020 until July 2020. Data were pulled from the electronic medical records at participating Northwestern Medicine hospitals. These were cross-referenced with the cancer center database, and a retrospective review was performed by the senior authors to extract additional clinical and laboratory data on patients with a history of malignancy and a documented SARS-CoV-2 infection. Baseline demographics, laboratory data, and clinical outcomes (death, cause and date of death, serious infections) were collected from electronic medical records. Serious COVID-19 infections were defined as infections leading to critical illness, i.e., admission to the intensive care unit (ICU) or the need for mechanical ventilatory support. Other clinical outcomes of interest were those related to the malignancy, such as type, early (I, II) versus advanced (≥III) stage, and treatment history (radiation, cytotoxic chemotherapy, immunotherapy, and timeframe in relation to COVID diagnosis and/or admission). Of note, patients who were diagnosed years ago at an early stage, their tumor surgically removed and then recurred as metastatic disease around the time of COVID-19 diagnosis, and therefore not treated, were lumped under “never treated” rather than “surgically treated”. In situ malignancies, basal cell carcinomas, and monoclonal gammopathy with unknown significance (MGUS) were excluded. Laboratory data at the time of admission or closest to COVID-19 diagnosis were collected and included blood counts and inflammatory markers. Patients were followed up for overall survival until 1 May 2021. All data were stored electronically as a REDCap database at Northwestern Memorial Hospital, and only approved members had access to its contents. The study was approved by the Institutional Review Board (IRB) board of Northwestern University.

### 2.2. Statistical Analysis

Descriptive statistics, including median (range) and mean for continuous variables, and count (percentage) for discrete variables were used to summarize baseline characteristics. Clotting event rates were compared by malignancy type (hematologic vs. solid tumors) using Pearson’s Chi-squared test. The method of Kaplan–Meier was used to estimate overall survival from the time of COVID-19 diagnosis to death or date of last follow-up. Logrank test was used to compare overall survival between groups, and hazard ratios (HRs) between groups were estimated via Cox proportional hazards models. Proportional hazard assumption was verified. Univariate logistic regression models were used to assess whether clinical predictors were associated with ICU admission or intubation, and the odds ratios (ORs) between groups were reported. Univariate Cox proportional hazards models were used to compare overall survival and to estimate HRs between groups based on laboratory predictors. Multivariable regression models were used to obtain estimates for clinical and laboratory predictors adjusting for age, race, and gender (Appendix A).

## 3. Results

### 3.1. Baseline Characteristics

Our analysis included 416 cancer patients with SARS-CoV-2 infection being managed for their malignancy between March 10th, 2020, and July 6th, 2020. Patients’ baseline characteristics are detailed in Table 1. The median age was 60, and the cohort included 224 (53.8%) females and 192 (46.2%) males; 288 (72.0%) patients were White, 70 (17.5%) were Black, and 12 (3.0%) were Asian. Most patients had solid tumors (*n* = 344, 82.7%), among whom the majority had early-stage disease (*n* = 263, 76.5%). The most common solid tumor malignancies were genitourinary (GU) (*n* = 79, 23.0%), breast (*n* = 68, 19.8%), gynecologic (*n* = 34, 9.9%), gastrointestinal (*n* = 32, 9.3%), and lung (*n* = 13, 3.8%). Among hematological malignancies, the most common types were lymphoproliferative disorders (*n* = 41, 56.9%), followed by multiple myeloma (*n* = 14, 19.4%), myeloid malignancies (*n* = 7, 9.7%), and myeloproliferative neoplasms (*n* = 5, 6.9%); 101 (24.3%) patients were undergoing cancer treatment at the time of their COVID-19 diagnosis, including cytotoxic chemotherapy without immunotherapy (*n* = 44, 10.6%), immunotherapy (*n* = 14, 3.4%) or radiation therapy (*n* = 11, 2.6%). An additional *n* = 1 (0.2%) and *n* = 6 (1.4%) patients completed chemotherapy within 3 and 3–12 months prior to COVID diagnosis, and *n* = 2 (0.5%) completed immunotherapy within 3–12 months (Appendix A).

### 3.2. Clinical Outcomes

#### 3.2.1. Clotting Complications

Clotting complications, defined as upper or lower extremity deep venous thromboses, pulmonary emboli (PE), cerebrovascular accidents, and myocardial infarctions, occurring within 14 days of COVID-19 diagnosis, took place in 43 (10%) of our cohort. Among patients with advanced solid tumors, 11 (16%) had a clotting event. Patients with early-stage tumors or hematological malignancies had comparable rates at 9.1% and 8.3%, respectively (Table 2).

#### 3.2.2. Clinical Predictors of Mortality

At the time of our analysis, 75 (18.0%) patients had died (Table 1), with (*n* = 46, 61.3%) deaths due to COVID-related complications, mainly acute respiratory distress syndrome (ARDS) driving respiratory failure. Twenty-three (30.7%) of all deaths were secondary to cancer-related complications and/or cancer progression. Overall, 30 d mortality rate was 10.0% (95% CI, 7.1–12.9%). Using the log-rank test, we assessed patients’ characteristics that were associated with overall survival. Our analysis (Figure 1; Table 2) showed that patient’s age greater than or equal to 60 years old was associated with worse overall survival (6-mo OS: 0.76 [95% CI, 0.70–0.82]), compared to those below 60 (6-mo OS: 0.90 [95% CI, 0.86–0.94]; *p* < 0.001; Figure 1a). In addition, male patients had significantly shorter overall survival (6-mo OS: 0.75 [95% CI, 0.69–0.82]) compared to female patients (6-mo OS: 0.89 [95% CI, 0.85–0.93]; *p* < 0.001; Figure 1b). No association was found between overall survival and race groups or ethnicity groups (Figure 1c–d). The type of malignancy (solid tumor or hematological) had no direct correlation with risk of death (Figure 1e). Patients with early-stage solid tumors (6-mo OS: 0.87 [95% CI, 0.83–0.91]), however, had a significantly higher overall survival than those with advanced-stage solid tumors (6-mo OS: 0.73 [95% CI, 0.62–0.85]; *p* = 0.002) or hematological malignancies (6-mo OS: 0.77 [95% CI, 0.68–0.88]; *p* = 0.009; Figure 1f; Table 2). Patients who were on treatment had worse overall survival (6-mo OS: 0.74 [95% CI, 0.68–0.84]) than those who were not (6-month OS: 0.85 [95% CI, 0.82–0.90]; *p* = 0.007; Figure 1g). One possible reason is that whether patients were on treatment was confounded with factors that affected the overall survival, such as cancer type, disease severity, etc. When further considering when cancer treatment was completed (Figure 1h), treatment completion <3 months (6-mo OS: 0.86 [95% CI, 0.69–1]) or within 3–12 months (6-mo OS: 0.81 [95% CI, 0.66–1]) was associated with lower but not significantly different overall survival compared to patients who completed treatment >12 months prior (6-mo OS: 0.87 [95% CI, 0.82–0.91]). Treatment type was significantly associated with overall survival (*p* < 0.001; Figure 1i), with patients on current immunotherapy having the lowest overall survival (6-mo OS: 0.43 [95% CI, 0.23–0.78]) compared to those who completed treatment at any time point prior to COVID diagnosis (HR = 4.89 [95% CI, 2.27–10.5]; *p* < 0.001). Current cytotoxic chemotherapy treatment (6-mo OS: 0.76 [95% CI, 0.64, 0.9]; Figure 1i) was marginally associated with lower overall survival (HR = 1.95 [95% CI, 1.00–3.83]; *p* = 0.051). Overall survival could not be estimated separately among patients who completed immunotherapy (*n* = 2) or chemotherapy (*n* = 7) within 1 year prior to COVID diagnosis due to small sample size. Association of these risk factors with overall survival adjusting for key demographic characteristics (age, gender, and race) was similar (Appendix A).

#### 3.2.3. Clinical Predictors of Critical Illness

We defined critical COVID-19 illness as infections leading to admission to the intensive care unit (ICU admission) or the requirement for mechanical ventilatory support (ICU intubation). A total number of 72 (17.6%) patients was admitted to ICU, and 43 (10.5%) patients needed ICU intubation. Our analysis showed that older age at diagnosis of COVID-19 was associated with an increased risk of ICU admission (OR, 1.03 [95% CI, 1.01–1.05] per year of age; *p* < 0.001) and ICU intubation (OR, 1.03 [95% CI, 1.00–1.05] per year of age; *p* = 0.023). Male patients also had higher rates of ICU admission (OR, 2.22 [95% CI, 1.32–3.79]; *p* = 0.003) and mechanical ventilation (OR, 2.16 [95% CI, 1.14–4.22]; *p* = 0.021) compared to female patients, as did those with hematological malignancies compared with patients with advanced stage or early-stage solid tumors (ICU admission: OR, 0.42 [95% CI, 0.24–0.78], *p* = 0.004; ICU intubation: OR, 0.34 [95% CI, 0.17–0.68], *p* = 0.002) (Table 3). Of note, race, ethnicity, and cancer-directed treatment completion history or treatment type were not associated with disease severity in our population. The association between severe illness and tumor type and stage remained statistically significant when adjusted for age, race and gender, whereas cancer treatment related variables were similarly not associated with ICU admission or intubation (see Appendix A).

#### 3.2.4. Laboratory Predictors of Mortality and Disease Severity

We also examined the relevance of certain laboratory values such as blood counts and inflammatory markers, obtained on day of admission for hospitalized patients or within 14-days of COVID-19 diagnosis for non-hospitalized patients (Figure 2). Absolute neutrophil count (ANC) had no significant association with overall survival (HR, 1.05 [95% CI, 0.38–2.86], *p* > 0.9) or need for ICU admission (OR, 1.43 [95% CI, 0.47, 6.23], *p* = 0.6). An absolute lymphocyte count (ALC) >1000 K/µL, compared to that lower than 500 K/µL, was found to be protective against death (HR, 0.40 [95% CI, 0.22–0.74], *p* = 0.003), ICU admission (OR, 0.35 [95% 0.17–0.75], *p* = 0.006), and need for intubation (OR, 0.27 [95% CI, 0.11–0.66], *p* = 0.004). A hemoglobin greater than 10 g/dL was also associated with decreased likelihood of death (overall *p* < 0.001), ICU admission (overall *p* < 0.001), and need for intubation (overall *p* = 0.006). An increase in platelet count level was associated with improved survival (overall *p* < 0.001) and decreased probability for ICU admissions (overall *p* < 0.001) or intubation (overall *p* = 0.008). Analysis of inflammatory markers demonstrated that a ferritin level between 500 and 1000 ng/mL compared to that below 500 (HR, 2.80 [95% CI, 1.39–5.64], *p* = 0.004), a lactic acid level ≥ 5 mmol/L compared to that ≤2 mmol/L (HR, 4.44 [1.59, 12.1], *p* = 0.004), D-dimer ≥500 (µg/mL) (HR, 2.27 [95% CI, 1.22–4.22], *p* = 0.009), and a procalcitonin ≥0.5 ng/mL (HR, 2.27 [95% CI, 1.30–3.97], *p* = 0.004) were associated with increased mortality. These same cutoffs were also associated with increased ICU admission for ferritin (OR, 2.53 [95% CI, 1.06–5.93], *p* = 0.034) and lactate (OR, 9.28 [95% CI, 1.33–184], *p* = 0.049) as did an LDH level ≥250 U/L (OR, 4.67 [95% CI, 2.25–10.4], *p* < 0.001) and a CRP ≥ 10 mg/dL (OR, 2.88 [95% CI, 1.40–5.96], *p* = 0.004). Finally, CRP (≥10) (OR, 2.47, [95% CI, 1.13, 5.39], *p* = 0.022) and LDH (≥ 250) (OR, 4.46 [95% CI, 1.92–11.7], *p* = 0.001) correlated with an increased need for intubation (Table 4). Association with ICU admission, ICU intubation, and overall survival was similar in multivariable regression models adjusted for age, race, and gender (Appendix A).

## 4. Discussion

We conducted an analysis of outcomes and predictive factors in COVID-19 infected cancer patients at a large hospital network located in one of the hardest hit cities in the US during the first wave. To our knowledge, this is the largest report of COVID-19 infections in cancer patients in the state of Illinois. Numerous hypotheses have been confirmed from our analysis. Patients with malignancy are at an increased risk of developing severe complications and dying from SARS-CoV-2 infection compared to the general population where critical illness rates were approximately 14 to 19% [17,19] and death rates 1 to 4% [15] at the start of the pandemic and before the advent of the COVID-19 vaccine. The COVID-19 and Cancer Consortium (CCC19) database reported a death rate of 13% among the COVID-19 infected cancer population [16] compared to our finding of 18.0%. A report of 334 cancer patients from Mount Sinai Health System found an 11% mortality rate [15], whereas a series of 218 patients from the Montefiore Health System reported a 28% rate [20]. Mehta et al. also reported an 11% intubation rate [15], almost identical to our 10.5%. Rates of admission to the ICU varied widely among the different studies. In a German study, it was reported to be as high as 36% [21]. A UK cancer center had rates as low as 7% [22], while the aforementioned Canadian/Spanish/American cohort reported an ICU admission rate of 14% [16]. Our data are consistent with these findings, with an observed rate of 17.3%. These differences could be explained by the fact that the pandemic hit different geographic locations at different timelines, and that the reported numbers are merely a snapshot taken at different points of the COVID-19 wave. These differences may also be largely driven by global health disparities and socioeconomical factors [15,23]. In our cohort, more severe disease and increased mortality was associated with male gender and advanced age. Although our findings were comparable to previously published data [15,24], Shoumariyeh et al. found no difference in mortality between cancer and non-cancer patients, infected with COVID-19, when adjusted for age, gender, and comorbidities [21]. Data, however, were limited to 39 patients in a single center. Our data are also compatible with Dai et al.’s findings that patients with advanced solid tumors had the worst outcomes, followed by hematologic malignancies, compared to those with early-stage disease [24]. Interestingly, there was no difference in mortality and infection severity between different cancer subtypes.

In accordance with Dai et al.’s findings [24], we also found that current immunotherapy treatment led to higher rates of death, although we were not able to assess outcomes among patients who completed immunotherapy within 1 year due to low numbers. Patients undergoing cytotoxic therapy also had worse overall survival, although less so than patients on immunotherapy. This finding fits the mechanism of action that has been suggested to lead to the clinical sequelae after a SARS-CoV2 infection. It is thought that the virus acts through hyperactivation of a diffuse inflammatory response, or “cytokine storm”. This leads to epithelial cell injury and death in numerous organs such as pneumocytes in lungs and podocytes in kidneys. ARDS and acute kidney injury (AKI) ensue [25,26]. It is therefore plausible that the use of immunotherapy in this subset of patients can lead to an exaggerated release of cytokines and an amplified inflammatory response, causing higher rates of ARDS, ICU admission, intubation, and death [27,28]. This cytokine storm has also been hypothesized to deplete the immune system, particularly of lymphocytes, suppress production of hematopoietic cells, and activate the coagulation cascade [26]. Hematologic findings from our cohort support this hypothesis, with higher platelet counts, ALC ≥ 1000, and hemoglobin ≥ 10 being protective against severe infection and death. Similarly, results from inflammatory marker analysis support the theory of “death by inflammation”, as ferritin ≥ 500, d-dimer ≥ 500, procalcitonin ≥ 5, CRP ≥ 10, and LDH ≥ 250 led to worse clinical outcomes.

The rapid spread of the COVID-19 pandemic has led to uncharted challenges facing the global healthcare system. Patients with malignancies have been one of the most at risk subgroups. Not only have they had higher rates of severe infections (ICU admission, intubation, and death), but in all reports they have consistently displayed higher rates of infection owing to their frequent hospital visits and long hours spent in virus-contaminated waiting rooms and infusion clinics [17,29]. Our study attempts to give a timely and accurate account of our experience in order to bring forth strategies to help alleviate the impact of this pandemic. Patients with advanced solid tumors and hematological malignancies seemed to be the most susceptible to severe infections and poor outcomes, as were those with advanced age. It is therefore imperative that they and their families be presented with this information when discussing their advanced directives. Conversely, the lack of association between mortality (or severe infections) and cytotoxic and radiation therapies is re-assuring. As described in our analysis, the ability of certain laboratory and inflammatory markers to consistently predict poorer outcomes also provides healthcare providers and patients with an added tool to be more proactive in their care, be it an earlier administration of oxygen, systemic steroids, or admission to units where vitals can be more closely monitored. Of course, the role of educating and vaccinating this vulnerable population cannot be understated. In fact, as our analysis preceded the development of currently available safe and effective vaccines, it is possible that the high-risk factors identified here will change for vaccinated patients.

Our study offers a significant amount of valuable data, but it is merely a snapshot of a cancer program experience in one of the epicenters of the pandemic. It is limited by the fact that it did not include noncancer patients as a comparative control to truly characterize the outcomes. In addition, it did not account for all Northwestern Medicine patients, as some were diagnosed and treated at outside facilities and local hospitals, especially those that had a relatively quiet and uneventful clinical course. This will undoubtedly select towards poorer outcomes. The clinical management of cancer patients with COVID-19 remains a challenge. Our experience will hopefully provide necessary data and much needed guidance to help tailor policy.

## Figures and Tables

**Figure 1 cancers-14-02209-f001:**
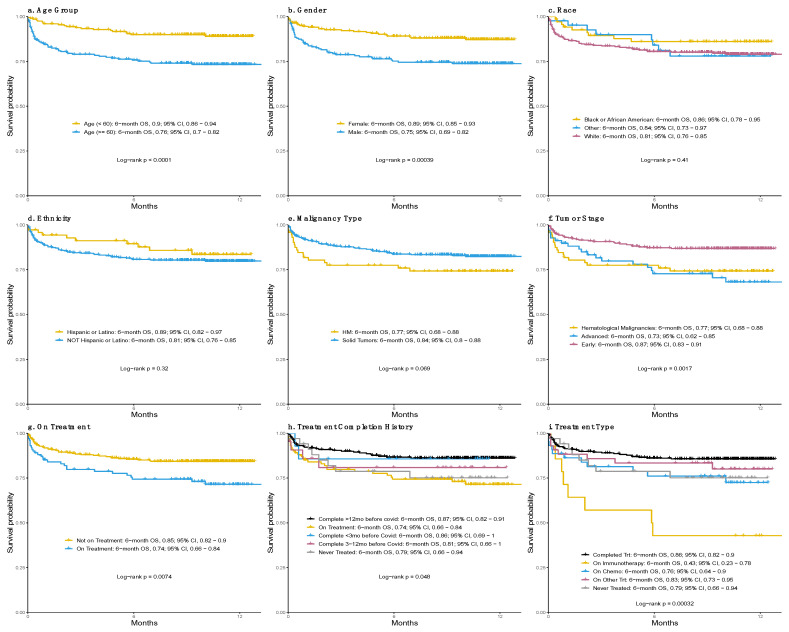
Overall survival by age group (**a**), gender (**b**), race (**c**), ethnicity (**d**), malignancy type (**e**), tumor stage (**f**), active treatment (**g**), timeline of treatment completion (**h**) and type of treatment (**i**).

**Figure 2 cancers-14-02209-f002:**
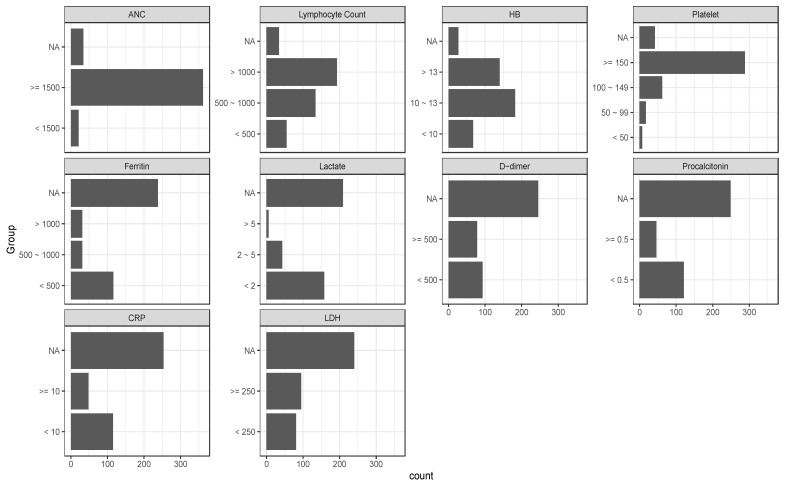
Distributions of lab results.

**Table 1 cancers-14-02209-t001:** Baseline demographics and outcomes.

Characteristic	*n* = 416 ^1^
Age	
Median (Minimum–Maximum)	60 (9–92)
Mean (SD)	59 (16)
Days Since COVID Diagnosis	
Median (Minimum–Maximum)	296 (0–462)
Mean (SD)	240 (128)
Gender	
Female	224 (53.8%)
Male	192 (46.2%)
Race	
Asian	12 (3.0%)
Black or African American	70 (17.5%)
White	288 (72.0%)
Other	30 (7.5%)
Unknown	16
Ethnicity	
Hispanic or Latino	71 (17.7%)
Not Hispanic or Latino	330 (82.3%)
Unknown	15
Malignancy Type	
HM	72 (17.3%)
Solid tumors	344 (82.7%)
Tumor Stage	
Low	263 (65.3%)
High	68 (16.9%)
HM	72 (17.9%)
Unknown	13
On Treatment	
No	312 (75.5%)
Yes	101 (24.5%)
Unknown	3
Treatment Completion History	
Complete >12mo before COVID	237 (58.1%)
On Treatment	101 (24.8%)
Complete <3mo before COVID	14 (3.4%)
Complete 3–12mo before COVID	22 (5.4%)
Never treated	34 (8.3%)
Unknown	8
Treatment Type	
Completed Trt	273 (66.9%)
On Immunotherapy	14 (3.4%)
On Chemo	44 (10.8%)
On Other Trt	43 (10.5%)
Never Treated	34 (8.3%)
Unknown	8
Clotting Events	
No	371 (89.6%)
Yes	43 (10.4%)
Unknown	2
Death	
No	341 (82.0%)
Yes	75 (18.0%)
Cause of Death (COVID)	
Alive	341 (82.0%)
COVID related	46 (11.1%)
Not COVID related	23 (5.5%)
Unknown	6 (1.4%)
ICU Care	
No	336 (82.4%)
Yes	72 (17.6%)
Unknown	8
ICU Intubation	
No	367 (89.5%)
Yes	43 (10.5%)
Unknown	6

^1^*n* (%).

**Table 2 cancers-14-02209-t002:** Clotting events in solid tumors vs. hematological malignancies.

Characteristic	No, *n* = 371 ^1^	Yes, *n* = 43 ^1^	*n*	*p*-Value ^2^
Solid tumors or HM			414	0.5
HM	66 (92%)	6 (8.3%)		
Solid tumors	305 (89%)	37 (11%)		
Stage			402	0.2
Early	239 (91%)	24 (9.1%)		
Advanced	56 (84%)	11 (16%)		
HM	66 (92%)	6 (8.3%)		
Unknown	10	2		

^1^*n* (%); ^2^ Pearson’s Chi-squared test excluding patients with unknown stage (excluding unknowns).

**Table 3 cancers-14-02209-t003:** Clinical predictors of ICU admission, ICU intubation, and overall survival.

Outcomes by Clinical Parameters	ICU Admission				ICU Intubation			Overall Survival		
Characteristic	*n*	OR ^1^	95% CI ^1^	*p*-Value	OR ^1^	95% CI ^1^	*p*-Value	HR ^1^	95% CI ^1^	*p*-Value
Age	408	1.03	1.01, 1.05	<0.001	1.03	1.00, 1.05	0.023	1.05	1.03, 1.07	<0.001
Gender										
Female	219	—	—		—	—		—	—	
Male	189	2.22	1.32, 3.79	0.003	2.16	1.14, 4.22	0.021	2.31	1.44, 3.73	<0.001
Race				0.11			0.2			0.4
White	284	—	—		—	—		—	—	
Black or African American	67	1.81	0.92, 3.42	0.074	1.83	0.80, 3.93	0.13	0.62	0.31, 1.26	0.2
Other	42	1.80	0.79, 3.83	0.14	1.74	0.61, 4.29	0.3	0.92	0.44, 1.93	0.8
Ethnicity										
Hispanic or Latino	70	—	—		—	—		—	—	
Not Hispanic or Latino	324	0.80	0.42, 1.59	0.5	0.75	0.35, 1.75	0.5	1.40	0.72, 2.72	0.3
Malignancy Type										
HM	71	—	—		—	—		—	—	
Solid tumors	337	0.42	0.24, 0.78	0.004	0.34	0.17, 0.68	0.002	0.61	0.36, 1.04	0.072
Tumor Stage				0.006			0.006			0.002
Low	260	—	—		—	—		—	—	
High	65	1.35	0.62, 2.77	0.4	1.23	0.43, 3.04	0.7	2.41	1.37, 4.24	0.002
HM	71	2.79	1.48, 5.19	0.001	3.24	1.54, 6.71	0.002	2.16	1.22, 3.84	0.009
On Treatment										
No	308	—	—		—	—		—	—	
Yes	98	1.53	0.86, 2.66	0.14	1.70	0.84, 3.34	0.13	1.89	1.18, 3.05	0.008
Treatment Completion History				0.6			0.6			0.057
Complete >12 mo before COVID	234	—	—		—	—		—	—	
On Treatment	98	1.54	0.84, 2.76	0.2	1.73	0.82, 3.53	0.14	2.16	1.29, 3.62	0.003
Complete <3 mo before COVID	14	1.45	0.32, 4.92	0.6	0.79	0.04, 4.26	0.8	1.09	0.26, 4.54	>0.9
Complete 3–12 mo before COVID	21	0.89	0.20, 2.79	0.9	1.02	0.16, 3.85	>0.9	1.55	0.55, 4.38	0.4
Never treated	34	0.92	0.30, 2.35	0.9	0.99	0.22, 3.09	>0.9	1.84	0.85, 4.01	0.12
Treatment Type				0.5			0.6			0.001
Completed Trt	269	—	—		—	—		—	—	
On Immunotherapy	13	1.58	0.34, 5.40	0.5	1.88	0.28, 7.54	0.4	4.89	2.27, 10.5	<0.001
On Chemo	44	1.17	0.48, 2.58	0.7	1.68	0.59, 4.14	0.3	1.95	1.00, 3.83	0.051
On Other Trt	41	1.93	0.87, 4.05	0.092	1.77	0.62, 4.40	0.2	1.38	0.64, 2.96	0.4
Never Treated	34	0.91	0.30, 2.29	0.8	1.00	0.23, 3.08	>0.9	1.76	0.82, 3.79	0.15

^1^ OR = Odds Ratio, CI = Confidence Interval. HR = Hazard Ratio.

**Table 4 cancers-14-02209-t004:** ICU admission, ICU intubation, and overall survival according to laboratory parameters.

Outcomes by Laboratory Parameters	ICU Admission				ICU Intubation			Overall Survival		
Characteristic	Summary Statistics ^1^	OR ^2^	95% CI ^2^	*p*-Value	OR ^2^	95% CI ^2^	*p*-Value	HR ^2^	95% CI ^2^	*p*-Value
ANC										
<1500	21 (5.5%)	—	—		—	—		—	—	
≥1500	361 (95%)	1.43	0.47, 6.23	0.6	NA	NA	NA	1.05	0.38, 2.86	>0.9
Unknown	34									
Lymphocyte Count				0.002			0.007			0.011
<500	55 (14%)	—	—		—	—		—	—	
500–1000	134 (35%)	0.93	0.46, 1.94	0.9	0.72	0.32, 1.67	0.4	0.66	0.36, 1.20	0.2
>1000	193 (51%)	0.35	0.17, 0.75	0.006	0.27	0.11, 0.66	0.004	0.40	0.22, 0.74	0.003
Unknown	34									
HB				<0.001			0.006			<0.001
<10	67 (17%)	—	—		—	—		—	—	
10–13	182 (47%)	0.31	0.16, 0.61	<0.001	0.33	0.15, 0.71	0.004	0.47	0.29, 0.78	0.003
>13	140 (36%)	0.32	0.16, 0.64	0.001	0.32	0.14, 0.72	0.007	0.13	0.06, 0.29	<0.001
Unknown	27									
Platelet				<0.001			0.008			<0.001
<50	7 (1.9%)	—	—		—	—		—	—	
50–99	17 (4.5%)	0.31	0.04, 1.93	0.2	1.36	0.21, 11.6	0.8	0.96	0.26, 3.64	>0.9
100–149	62 (17%)	0.10	0.01, 0.51	0.010	0.22	0.04, 1.83	0.12	0.34	0.10, 1.20	0.093
≥150	288 (77%)	0.07	0.01, 0.33	0.002	0.25	0.05, 1.83	0.11	0.23	0.07, 0.73	0.013
Unknown	42									
Ferritin				0.064			0.3			0.010
<500	116 (65%)	—	—		—	—		—	—	
500–1000	31 (17%)	2.53	1.06, 5.93	0.034	1.94	0.68, 5.17	0.2	2.80	1.39, 5.64	0.004
>1000	31 (17%)	2.00	0.80, 4.80	0.13	1.94	0.68, 5.17	0.2	2.13	1.00, 4.55	0.051
Unknown	238									
Lactate				0.14			0.7			0.018
<2	158 (76%)	—	—		—	—		—	—	
2–5	43 (21%)	1.08	0.50, 2.23	0.8	1.04	0.41, 2.39	>0.9	1.17	0.63, 2.19	0.6
>5	6 (2.9%)	9.28	1.33, 184	0.049	2.27	0.30, 12.2	0.4	4.44	1.59, 12.4	0.004
Unknown	209									
D-dimer										
< 500	93 (54%)	—	—		—	—		—	—	
≥500	78 (46%)	1.48	0.77, 2.89	0.2	1.53	0.73, 3.26	0.3	2.27	1.22, 4.22	0.009
Unknown	245									
Procalcitonin										
<0.5	121 (72%)	—	—		—	—		—	—	
≥0.5	46 (28%)	1.52	0.73, 3.11	0.3	1.54	0.65, 3.49	0.3	2.27	1.30, 3.97	0.004
Unknown	249									
CRP										
<10	115 (71%)	—	—		—	—		—	—	
≥10	48 (29%)	2.88	1.40, 5.96	0.004	2.47	1.13, 5.39	0.022	1.25	0.66, 2.35	0.5
Unknown	253									
LDH										
<250	81 (46%)	—	—		—	—		—	—	
≥250	95 (54%)	4.67	2.25, 10.4	<0.001	4.46	1.92, 11.7	0.001	1.29	0.70, 2.36	0.4
Unknown	240									

^1^*n* (%); ^2^ OR = Odds Ratio, CI = Confidence Interval, HR = Hazard Ratio.

## Data Availability

The data presented in this study are available on request from the corresponding author. The data are not publicly available due to HIPAA considerations.

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
