# Peer review of "Outcomes of Cancer Patients with COVID-19 in a Hospital System in the Chicago Metropolitan Area"

_cancers, 2022, doi:10.3390/cancers14092209_

Round 1

Reviewer 1 Report

This is an excellent and meticulously documented retrospective cohort study of the relationship between malignancies and severe COViD-19 outcomes including ICU admission.

introduction

“Data generated from the 52 COVID-19 and Cancer Consortium (CCC19) also established that patients with a history of cancer had a more severe COVID-19 illness and a higher fatality rate compared to that of the general population [16]. Patients with malignancy are also more likely to get in-fected with SARS-CoV-2 owing to the nosocomial transmission of COVID-19 and the need of cancer patients for frequent hospital visits and regular contact with virus-contaminated areas [17, 18].”

[please provide more details of the design of these studies (retrpspective cohort, prospective cohort?), statistical controls for variables likely to corelate with outcomes, such as comorbidities, age, gender, ethnicity, therapies …]  

Statistical analysis

The method of Kaplan-Meier was used to esti-90 mate overall survival from the time of COVID-19 diagnosis to death or date of last follow-91 up, and the log rank analysis was used to compare overall survival between groups, and 92 hazard ratios (HR’s) between groups were estimated via Cox proportional hazards mod-93 els. Univariate logistic regression models were used to assess whether clinical predictors 94 were associated with ICU admission and intubation, and the odds ratios (OR’s) between 95 groups were reported. Univariate Cox proportional hazards models were used to com-96 pare overall survival and to estimate HR’s between groups based on laboratory predic-97 tors.

[You have recorded multiple variable likely to be associated with poorer outcomes. Importantly, can you please perform multiple linear regression using data for which you have adequate numbers please? You are endeavouring to provide clinical care guidelines, and in the event that your data can be formulated in terms of a scoring system this will be important]

Conclusions

You reached several important conclusions and compared them to prior studies:

“Patients with malignancy are at an increased risk of developing severe 209 complications and dying from SARS-CoV-2 infection compared to the general population 210 where critical illness rates were approximately 14 to 19% [17, 19] and death rates 1 to 4% 211 [15], at the start of the pandemic and before the advent of the COVID-19 vaccine.”

“Rates 217 of admission to the ICU varied widely among the different studies. In a German study, it 218 was reported to be as high as 36% [22]. A UK cancer center had rates as low as 7% [23] 219 while the aforementioned, Canadian/Spanish/American cohort, reported an ICU admis-220 sion rate of 14% [16]. Our data are consistent with these findings, with an observed rate 221 of 17.6%. These differences could be explained by the fact that the pandemic hit different 222 geographic locations at different timelines and that the reported numbers are merely a 223 snapshot taken at different points of the COVID-19 wave. These differences may also be 224 largely driven by global health disparities and socioeconomical factors [20, 24]. In our co-225 hort, more severe disease and increased mortality was associated with male gender and 226 advanced age.”

“we also found that Immunotherapy treat-235 ment, within 3 months or a year of COVID-19 diagnosis, led to a more severe disease and 236 higher rates of death. Oddly, this did not apply to patients undergoing cytotoxic therapy 237 (or radiation) within 3 months or a year of their diagnosis. This finding fits the mechanism 238 of action that has been suggested to lead to the clinical sequelae after a SARS-CoV2 infec-239 tion. It is thought that the virus acts through hyperactivation of a diffuse inflammatory 240 response, or “cytokine storm”. This leads to epithelial cell injury and death in numerous 241 organs such as pneumocytes in lungs and podocytes in kidneys. ARDS and acute kidney 242 injury (AKI) ensue [26, 27]. It is therefore plausible that the use of Immunotherapy in this 243 subset of patients, can lead to an exaggerated release of cytokines and an amplified in-244 flammatory response, causing higher rates of ARDS, ICU admission, intubation and death 245 [28, 29]. This cytokine storm has also been hypothesized to deplete the immune system, 246 particularly of lymphocytes, suppress production of hematopoietic cells and activate …”

[Are you able through multiple regression with adjustments for as many relevant data items for which you have adequate numbers to refine your data into a predictive model, to which vaccination data on vaccinated patients could later be added and compared to outcomes from other centers?]

Author Response

Please find my responses to reviewers attached

Reviewer 2 Report

Proposed paper is interesting and well written. However, some revisions are needed before pubblication:

  • A clotting complication of 10% is showed. How much it is in the years previous COVID in the same hospital? please use an historical cohort as controls in order to understand this point.
  • Similarly also data on previous mortality rate are needed. On the mortality the very numerous analysis on K-M curves present in table 1 are unusefull. A single multivariable model with all the factor implicated demonstrating which one are predictors of mortality is needed. So treat mortality similarly to what have been done for ICU admisino/ventilation.
  • Once again all the analysis done regarding biochemical parameter are not needed. Please putt biochemical parameter into the models and see which one are predictors.
  • Summarizing, after the first descriptive part three models with all the significantly associated parameter for mortality, ICU admission and ICU ventilation should be done.

Author Response

(The authors gave the same response as above.)

Round 2

Reviewer 1 Report

Thanks to the authors who have been able to make some small changes within the limits of their data availability.

Reviewer 2 Report

Authors replies to all the query raised and paper improves and can now be accepted for pubblication.